# Towards a HR Framework for Developing a Health-Promoting Performance Culture at Work: A Norwegian Health Care Management Case Study

**DOI:** 10.3390/ijerph17249164

**Published:** 2020-12-08

**Authors:** Rune Bjerke

**Affiliations:** Department of Leadership and Organization, Kristiania University College, 0107 Oslo, Norway; rune.bjerke@kristiania.no

**Keywords:** health-promoting performance culture, HR-strategy, organizational drivers, health care management, employee mental and physical health

## Abstract

The Norwegian Institute of Public Health (NIPH) states that Norway faces several major health challenges. Sick leave is at 6% and costs employers approximately EUR 1.75 billion annually. The NIPH proposes, with the support of the Public Health Act and the national strategy HealthCare21, that preventive measures should be developed to address negative lifestyle factors in order to decrease the number of new cases in the related disease groups (e.g., stroke, high blood pressure, type 2 diabetes, osteoporosis, obesity). The purpose of this article is to answer why and how organisations should develop a health-promoting performance culture and to provide a conceptual model displaying the importance of this type of culture for organisational performance. To boost the national health standard as a consequence of employee physical activity at work, I suggest additional occupational safety and health (OSH) directives. Based on cross-disciplinary theorizing, I propose a definition of a health-promoting performance culture. This kind of culture consists of dimensions such as health objectives, shared health values, supportive health environment, goal-oriented and value-based behaviour of leaders and employees, and a winning mindset. In addition, the article underscores the importance of related individual HR drivers like fun at work, engagement, physical and mental health for increasing organisational performance. The company cases used in this paper, Schibsted, Gjensidige, Findus and Wilhelmsen, and findings from five in-depth interviews, indicate that health-promoting activities are the result of either an HR strategy or individuals’ initiative and voluntariness among the companies’ sports enthusiasts. The case of Findus exemplifies an ongoing development toward a health-promoting performance culture and the importance of leaders’ participation. The findings support several elements of the conceptual model showing the relations between a health-promoting performance culture, individual HR drivers and organisational performance. A framework for developing a health-promoting performance culture in practice is presented.

## 1. Introduction

The total health spending for the Norwegian population for 2019 is estimated to be EUR 35.7 billion, EUR 6250 per person. Treatment costs account for more than 25% of the state budget of EUR 132.4 billion, although only 2.5% was used for preventive health measures [1]. NIPH proposes that preventive measures should be directed at life-shortening life factors [2]. The Public Health Act, which came into force on 1 January 2012, stipulates that all sectors have a responsibility for public health (the health sector and all levels of government). From the literature on health-promoting management, it is suggested that a number of methods can be used to increase employees’ awareness of the importance of physical activity in the workplace. Organisations have an opportunity to reduce sick leave and establish a supportive culture for facilitating physical activities for employees and a health-promoting environment [3,4]. One argument for developing a health-promoting performance culture is that physical activity has been shown to protect against several diseases and conditions such as cardiovascular disease, high blood pressure, diabetes 2 and obesity [4]. Anderson et al. [5]. discovered that it is possible to lower the weight levels of employees by changing employees’ behavior (physical activity, food choices and diet). Besides, Kuoppala, Lamminpää, and Husman [6] concluded that work health promotion is valuable for employees’ well-being, work ability and productivity in terms of fewer sickness-related absences. They claimed that activities involving exercise, lifestyle, and an ergonomics approach can be effective. Losina et al. [7] found that less employee physical activity is associated with more sickness-related absences from work, and they suggested that physical promotion interventions at work can contribute to decreasing unplanned sick leave.

Based on the health challenges in Norway and the authorities’ intention to improve public health through preventive measures, this conceptual and exploratory study uses relevant theory and empirical data to investigate what organizations can do to create a health-promoting organization. The purpose of the paper is twofold. First, to theorise about culture, individual HR drivers, and performance and explore how some Norwegian employers facilitate employee physical activity in the workplace. Second, to recommend how employers can contribute to improving both employees’ health and thus, at an accumulative level, the public health, through the development of a health-promoting performance culture. Many health problems in Norway result in sick leave and lost person-years [2,8]. Approximately 54,000 lost person-years for private business in 2017 (corresponding to cost of ca EUR 3.5 billion) can largely be explained by the short-term sick leave of 1–2 days [9]. Life-shortening lifestyle factors like tobacco use, high alcohol intake, unhealthy diet and physical inactivity, and the disease groups cancer, cardiovascular disease, chronic lung disease and diabetes, may be linked to sickness absence, which is approximately 6% [10]. In addition, stress negatively affects employees’ physical and mental health, which leads to organisations and society being financially burdened [11]. Several studies have addressed different sources of work stress [12]. These can be long working days, overtime, time pressure, and unclear roles [13]. 

Health has been defined by the World Health Organization as the presence of physical, mental and social well-being [14]. There are documented benefits of healthy employees. For instance, physical exercise can reduce sick leave and can support the development of an organisational health culture [15], and health-promoting measures can have a positive effect on performance [16]. Caverley, Cunningham, and MacGregor [17] suggested that activities to enforce workplace health can lead to improved presenteeism, and Kahn [18] maintained that loss of physical energy (one of four employee troubles) can influence employees´ psychological presence. Given that social interaction, close relationships, enough physical activity, and normal blood pressure are life-prolonging factors [19]. Organisations can work with preventive health measures and the life-prolonging factors to improve the mental and physical health of the 2.7 million people employed in Norway [10]. There is a considerable amount of research on employee wellness programs [20,21,22,23,24,25].

The contribution of this study is to explore how such programs are organised and implemented in Norway in a health-promoting performance culture perspective. By examining the Norwegian company cases, Findus, Schibsted, Gjensidige, and Wilhemsen, I first describe how some Norwegian companies organise the facilitation of health-promoting activities. Second, this article proposes a new concept of health-promoting performance culture integrating both a health culture and a performance culture. In the theory section below, the constructs of health-promoting performance culture and individual HR drivers as forces of organisational performance are discussed, and a conceptual model is proposed. Then, materials and method, results and discussion are presented. Finally, I conclude and propose a strategic HR framework, suggesting how to develop a health-promoting performance culture which is new to the health care management literature.

## 2. Theory

### 2.1. Organisational Culture and Health-Promoting Performance Culture

Below, I present the concepts and relationship between the central theoretical terms of this study: *organisational culture*, *health-promoting performance culture*, *employee competence*, *employee drivers* (e.g., commitment, mental and physical health), and *participatory leadership* and argue for the HR-related drivers of organisational performance (see Figure 1).

Organisational culture can be understood as the organisation’s rituals and symbols, formed and maintained by groups of employees who, together, shape the organisation over time. Culture can influence and be influenced by individuals. It can be seen as a shared memory shaped by past experiences and governing future action. The shared memory is strengthened and passed on through dialogue, conversation and storytelling [26,27,28]. It can be challenging to both change and govern [29,30]. According to Denison [31,32], the organisational culture consists of four measurable dimensions: vision/mission/values, adaptability, involvement, and consistency. A positive organisational culture will have a positive impact on the company’s financial performance, efficiency and profitability [31,33,34] and is a driver of the organisation’s performance, of which employee involvement has an important mediating influence [35]. Schein [29] argued that organisational culture consists of artefacts, values, norms and leaders’ basic assumptions (e.g., truths/beliefs about a way of solving certain tasks). These assumptions, often communicated by clear, persuasive leaders, can establish themselves as a common mindset and become part of the organisational culture. Artefacts can be offices, art, language, dress code, stories, ceremonies, rituals, and behaviours [28,29].

Organisational culture has also been documented to have a significant positive effect on individual performance. Nevertheless, employee participation is the most important explanatory variable for goal achievement of the organization [36]. Participation also includes leaders through participatory leadership (see below). For example, collaboration between departments is a sign of cross-functional coordination and is seen as a driving force for the development of competitive advantage and improved performance, and is linked to organisational culture [37]. Values are important for the organisation’s identity and culture [38], and values are provided by people [39]. Based on the organisation’s values, employees must perform by delivering good products, services and experiences, so that they and the organisation will be associated with positive brand associations and be preferred over other alternatives [38,40,41].

A health-promoting performance culture is a merger between a high-performance culture and a health culture. To define this construct, it is necessary to explain and define performance and health culture. Kaliprasad [42] described performance culture as the organisation’s ability to deliver results that are among the absolute best in the industry through success factors such as employees’ competence and continuous education, effective work groups, and leadership. According to Spector and Lane [43], a high-performance culture can be regarded as a driver of sustained high performance where shared values such as transparency, accountability and dialogue aid in shaping employee behaviour. Referring to the General Electric case (GE), a few of the important values were self-confidence, openness, and being fair [43]. In addition, the fit between promoted and practiced values was vital. de Waal [44] argued that the culture of a high-performance organisation is characterised by a winning mindset, high excellence in whatever the organisation does, shared meaningful core values and identity, a sense of community, transparency, openness, trust, empowerment, freedom, and autonomy with behavioural restrictions. Thus, a definition of a performance culture is suggested to be as follows (see also Figure 1).

A performance culture is characterised by collective excellence in thinking and acting based on shared core values, and goal-oriented, value-based managerial and employee behaviour that continuously inspires the development of transparency, trust, empowerment and autonomy as a way to foster a winning mindset and a sense of community.

Lin and Lin [45] argued that physical health of organisational members is a core, shared value representing an organisational health culture, which is disseminated formally or informally to the employees. In discussing the two constructs of organisational culture and climate, Zohar and Hofmann [46] suggest that organisational climate is a bottom-up energy flowing from employees reflecting the core values and assumptions, which shape the organisation’s culture. Allen and Allen [47] maintain that a sense of community, a shared vision, and a positive outlook are dimensions of a cultural climate. A safety climate or culture are seen as important outcomes of health and safety initiatives [48]. More specifically, Gurt and Elke [49] documented that culture, in terms of safety, is viewed as a mediator of health and safety leadership aiming at improving employee safety performance and health. They concluded that leadership has a strong effect on constructing and developing a corporate health culture.

Golaszewski et al. [15] suggest that a health culture is a social and organisational construction based on a collective set of core values, assumptions, and expectations guiding how employees think, feel, and behave in order to improve personal and employee health. Golaszewski, Allen and Edington [50] underline the importance of cultural (e.g., values, norms, physical touch points, and rituals) and structural factors (e.g., facilities, service, and administrative support) of an organisational health environment. Hoebbel et al. [51] discuss the Life-gain Health Culture Audit (LHCA) instrument which is intended to aid health promotion planning and evaluation (measure health culture) and includes factors such as norms, values, social support, touch points, and climate. McLeroy et al. [52] proposed an ecological model reflecting both individual and social factors as targets for health promotion interventions. The factors were: (1) Intrapersonal (knowledge, attitude, behaviour, skills, etc.); (2) Interpersonal (social, networks, social support systems, etc.); (3) Institutional (organisational characteristics, rules, and regulations); (4) Community (relations between organisations and institutions); (5) Public policy (local, state, and national laws and policy). These factors are regarded here as belonging to the supportive environment. Therefore, I propose the following definition of a health culture as a type of organisational culture, which is integrated in Figure 1:
A health culture is an inspiring ecosystem with shared health objectives, norms, and health values, which are supported by participative leadership, a supportive environment such as physical facilities and common employee touch points stimulating employee physical activity and sense of community to foster employee physical and mental health.

As a health-promoting performance culture is suggested to constitute both a performance and health culture, I propose the following definition of a health-promoting performance culture:
A type of organisational culture that is characterised by participative leadership, a winning mindset, value-based managerial and employee behaviour, norms, rituals, a sense of community, and a supportive health environment that contributes to achieving ambitious health and social goals, so that employees can perform to the highest levels, without mental or physical strain.

Based on the theorising above, it is of great interest to explore whether there is evidence of a health-promoting performance culture among some Norwegian companies. These companies have a reputation for facilitating employee physical activity, and thus may promote health issues among employees.

RQ1: Are there any indications of a health-promoting performance culture among the Norwegian companies?

In order to develop a performance culture, Sullivan [53] explains how the HR department must take responsibility for what is defined as strategic in the HR function and develop a competitive advantage in the management of human resources. The HR strategy must be future-oriented consisting of (competence or career) development programs for employees which enable them to achieve short-term and long-term goals. For example, wage strategies and strategies for creating meaningful content in employees’ job tasks have a positive influence on commitment and job satisfaction [54], which can be linked to improved health and well-being [55]. Albrecht et al. [56] maintain that mental and physical health are linked to employee engagement, organisational culture, and job satisfaction. Employee socialisation and performance management, of which employee competence is a part [42], also enhance employee engagement, which may result in a stronger competitive advantage. Therefore, the theory suggests that there is a relationship between employee competence and employee drivers (see Figure 1). 

*Employee competence* covers areas where knowledge, skills and experience affect an organisation’s performance [57]. At the company level, the composition of the company’s total experience, education, knowledge and skills is vital in order to achieve a competitive advantage [58]. Employee competence consists of skills, judgment and intelligence at the individual level [59] and knowledge and experience [60] (see Figure 1). Learning and development have a positive impact on employee well-being, productivity and profitability [61]. Through the development and change processes, these individual resources are transformed into a collective competence that can be linked to company performance [57]. Even though the employees’ individual competencies must be developed and nurtured, it could be argued that it is the collective power in an organisational culture that is the main driver of a health-promoting performance culture.

*Employee drivers* consist of several factors that affect the individual’s behaviour and performance at work. Mental health is considered here to be a very important factor. As mentioned, health is about the presence of physical, mental and social well-being [14]. Downward and Rasciute [62] found that there is a positive relationship between sports activities and the feeling of well-being, and in addition, impairment of physical energy can affect employees’ mental states [18]. Price, Bray, and Brown [63] conducted a study on workplace canteens and found that ‘there was openness to technical solutions such as smartphone apps, which not only solve the problem of information overload but also increase engagement with the customer’. Employees´ well-being can be divided into three constituents: subjective well-being, work well-being, and psychological well-being [64]. Satisfaction with life and general feeling of happiness constitute subjective well-being, and work well-being consists of job satisfaction and a feeling of happiness for the job [65]. Job satisfaction is defined as an emotional response that is felt when we experience psychological growth through achievement, recognition, work content, responsibility and advancement [66]. Self-acceptance, positive relationships with others, mastery of the environment, autonomy, personal growth, and the meaning of life are elements which constitute psychological well-being [67]. Positive correlations between job satisfaction, motivation and performance have also been documented [68]. Therefore, mental health, comprising these three well-being components, is defined as an employee driver (see Figure 1).

A study on the topic of fun at work, defined here as an employee driver, showed that fun activities (competitions, social events, team building and celebrations of work performance) can have a beneficial effect on employee performance. However, the ‘packing of fun’ must be compatible with the organisation’s strategy, results and employee competence [69]. In another study, fun at work was divided into the three categories: fun activities; socialisation among colleagues; and the management’s facilitation of fun and enjoyment. One of the conclusions of the study was that fun in the workplace has a reinforcing effect on employee relationships [70]. In addition, fun at work has a positive impact on job satisfaction [71]. Loehr [55] concludes that the benefits of employee engagement are enthusiasm, greater value contribution to the employer, and improved physical health and well-being. Drivers of employee engagement are defined by Bedarkar and Pandita [72] as communication, work–life balance and leadership. Employee engagement can be created through the alignment between leadership, job content and individual competence [73]. Bakker and Demerouti [74] concluded that committed employees are more creative, productive and prepared to make an extra effort. Work commitment is positively related to job satisfaction, organisational commitment and good health, and a health culture can be established by developing an organisational health environment [50]. Results in the study by Ford et al. [16] show that poor health is associated with significant reductions in work performance. Therefore, they propose interventions to improve health that can have a positive impact on performance. Figure 1 displays the employee-drivers of fun at work, job satisfaction, motivation, engagement, commitment, physical and mental health.

*Participatory leadership* is the driver which is responsible for the developments of the organisational culture through the communication of vision, values and goals, representing clear guidelines for employees [29,75]. From the theory on performance culture employee competence, education, and leadership emerge as critical factors for developing such a culture [42]. Thus, leaders must take responsibility for developing culture and individual HR drivers, shaping a health-promoting performance culture (see Figure 1). Hoert, Herd, and Hambrick [24] found that employees who recognised greater leader support for health promotion were involved in more wellness activities and positive health behaviours and they experienced less job stress. Ind and Bjerke [38,40] emphasise that leaders should be aware of such an interconnection challenge and work concurrently with the drivers to create a collective force toward goal achievements. They argue that leaders should create enthusiasm for processes that create new insight, energy and perseverance, while at the same time encouraging intuition and sensitivity to the needs of others. Managers should be physically present and perceived as part of the organisation and the employee community so that all employees stand together in the process of ‘living the brand’ [41]. This creates the conditions for the organisation to appear as a cohesive unit that delivers consistent messages, promises and product and service experiences to customers and other interest groups [38,40]. Therefore, participatory leadership can be a type of leadership that is useful for developing a health-promoting performance culture. The conceptual model as displayed in Figure 1, and the theorising above suggest that health and performance culture are types of organisational culture. I have explained and defined three types of cultures, and thus argued that a health-promoting performance culture is a hybrid of a health and a performance culture. Further, research shows that there are connections between culture and employee drivers. Based on related interdisciplinary theories, I also argue that participative leadership is required to develop both employee competence and employee drivers, enabling the creation of a health-promoting performance culture which subsequently improves organisational performance. Therefore, I maintain that if the employees’ mental and physical health are not well taken care of, the development of a health-promoting performance culture is slowed down, and this will affect performance. Based on such a conceptual model, it is interesting to ask questions about how and why some Norwegian companies facilitate health-promoting measures for employees.

RQ2: How and why do companies facilitate physical activity in the workplace?

### 2.2. Facilitation of Physical Activities in the Workplace

Crespo et al. [76] examined physical health-promoting activities in the workplace among adults who worked and lived outside the home (commuters). They concluded that a number of strategies for promoting physical activity in the workplace can improve the physical activity of employees. Brynjolfsson, Hitt and Yang [77] emphasised that health-promoting projects need the managers’ attention and documented that an employee portal/health portal improved communication about health-related content. Golaszewski, Allen, and Edington [50] stressed the importance of a supportive environment (work factors, cultural factors, and structural factors), and also a portal, a supportive structural factor, was argued to have an effect on employee health. They developed a conceptual model related to health promotion programs in the workplace and determined also that supporting structural factors were the most important. A separate health portal had a motivating influence on employees, which in turn contributed positively to employees’ health. Bjerke and Elvekrok [78] concluded that the health portal of Aker ASA, which sponsored the national ski team, akerkativ.no, had a significant impact on employees’ motivation to engage in physical activity.

Giddens, Leidner, and Gonzalez [79] found that pedometers, as support in a wellness program in the workplace, have a positive impact on employee health and well-being, indicating a positive relationship between the use of health technology and health services. Three main predictors of satisfaction with web-based training are suggested: (1) instructional design effectiveness; (2) website usability; (3) course usefulness [80]. An integration and flow between social, technical and managerial elements in a corporate portal is optimal. For instance, two of four technical factors are usability and effective information [81]. A positive attitude towards web-based health portals may be achieved when using sporting celebrities to mediate the messages [82]. Other elements like informational messages (physical activity and healthy eating) may be included in portals to influence employee health knowledge and attitudes [5]. Based on the connection between diet and health, Price, Bray, and Brown [63] conducted a study on canteen food and information about food in a workplace. They concluded that employees were open to technical solutions (mobile app) that could provide information about nutrition and lead to engagement in healthy eating. This means that the content and technological support tools in health promotion programs should be aimed at employees. It is therefore interesting to investigate which facilities and conditions Norwegian companies offer in the promotion of physical activity in the workplace. Thus, the last research question is as follows:

RQ3: What kind of facilities and structural factors are offered to employees to encourage physical activity?

The method is described below and outlines how the three research questions are answered.

## 3. Materials and Method

### 3.1. Design, Data Collection and Analysis

To answer the research questions, a multiple-case design [83] with four Norwegian cases Schibsted, Findus, Gjensidige and Wilhemsen was chosen. This design enables a comparative approach to investigate possible differences/similarities between the cases [84]. These companies were identified to be offering employee physical activity at work, health-promoting programs, facilities, and support systems, which were the objects to be analysed [85]. The logic behind an inductive approach was applied, and qualitative data were used to enrich the meaning of descriptions and emotions expressed in connection with the questions about the meaning of the facilitation for physical activity [83]. To find answers to the three research questions, a triangulation method with interviews and cases was applied to strengthen the credibility and reliability of the study [86,87]. In order to meet the reliability criterion [87,88] and to develop a credible representation of the employee health care programs, theory and findings from both documents of the four companies and the five different informants were used.

Based on the literature review, a semi-structured interview guide was developed. The guide was used to capture different themes (see below), descriptions and specific answers [89]. The interviews were conducted ‘online’ (technology from Zoom.us), with a selection of managers [90] who were responsible for the companies’ physical activity programs and HR functions. The informants were: (1) an event and office manager; (2) a head of people and group functions, (3) a head of facility management and real estate; (4) an HR director; (5) a business manager sales. The interviews lasted 60 min or longer. They were recorded using Zoom technology and then transcribed to ensure reliability [91,92]. The in-depth interviews were followed up by communication via e-mail to obtain additional information from the informants and to obtain answers to unresolved questions as a validation procedure [90,93]. The themes/codes that formed the basis for the comparison were health culture and performance culture, HR strategy, leaders´ participation, organisation, programs and employee participation, offers/benefits, facilities, programs, and communication/health technology. These were used as predefined codes for the analysis of the transcripts [91] to strengthen the validation [45]. During the analysis of the data, similarities/differences, descriptions and statements that could be linked to the pre-defined codes were searched for [89,94].

The companies provided examples of information material such as programs, intranet pages, and timetables of physical activities, and these were analysed as secondary data [85]. The same codes as listed above were used to continue the thematic analysis, as suggested by Braun, Clarke and Weate [95]. Document analysis is useful because documents are stable, and they describe a number of activities over a long period of time [92]. The combination of document analysis with analyses of interviews strengthened the credibility and reliability of the multiple-case study [86,87]. This study is theorised and completed within an interdisciplinary and exploratory perspective and is limited to four Norwegian cases, which precludes generalisations. However, conceptually, one of the goals was to develop, legitimise and explain theoretical contexts [96].

### 3.2. The Cases

#### 3.2.1. Findus

Findus is part of Nomad Foods, which is Europe’s largest supplier of frozen food. The company with over 4600 employees, an owner of 13 different production sites and a distributor of food to 17 countries, is 2.54 times larger than its largest competitor. The group has annual sales of 2.27 billion euros. The market share in Norway with the brand ‘Findus’ is 36.8%. The company’s vision and value wheel highlight an organisational culture that is characterised by innovation, change and development, effective management and brand building. ‘At Findus, we work to help families eat a little healthier and smarter every day - and we believe that this mentality starts in the workplace. We consider health and well-being as an important starting point for success and therefore, we launched our very own health and wellness program last year’, the informant states. The main goal is to stress that physical health is an important topic throughout the year, and the program includes topics such as physical activity, nutrition, mental health and sleep [97].

The wellness program ‘Our Well Way’ implemented by Findus, Norway, includes group activities, run by employees or by hired external instructors, and internal knowledge-promoting seminars with three focus areas. In January–April, the theme is ‘nutrition and good food’, May–August has ‘coping with stress and mental health’ on the agenda, and in September–December, the theme is ‘wellness and happiness’. The ‘Our Well Way’ program also serves as the company’s main platform is used for informing and raising awareness of the company’s core health-related values. The company offers flexible working hours and allows employees to train during working hours. Findus did not have its own arenas for physical activity, so employees had to move chairs and tables in the canteen to carry out training sessions. During the summer of 2020, the company’s head office moved into new premises, where the company shares fitness facilities with other companies (fitness room, spinning room and bicycle parking) in addition to having its own verandas to be used for yoga sessions. Findus covers employees’ external fitness centre membership fee with up to Nok 200 monthly and has insurance schemes covering injuries, physiotherapy and chiropractors. Findus has used a simple excel sheet for registration in physical group sessions and the intranet for distribution of information. 

#### 3.2.2. Schibsted

Schibsted is an international company that owns and manages several well-known digital consumer brands. The group employs more than 5000 employees (Norway, Denmark, Finland, Sweden and Poland). In Norway, Schibsted owns four of Norway’s ten largest newspapers; VG, Aftenposten, Bergens Tidene and Stavanger Aftenblad (SNL). Schibsted has implemented the four core values of integrity, innovation, unity and competitiveness. In one of Schibsted’s premises in Akersgata 55, the company has 1200 employees. This office space has a large fitness centre on the ground floor (basement), which includes two activity rooms. One room is designed for yoga and spinning, or other activities if the employees so wish.

Schibsted has a corporate agreement with the fitness centre SATS (chain). Hired instructors hold training sessions several times a week, and employees can participate for free. They receive health insurance, and access to an associated App that can be used to book a doctor’s appointment, physiotherapist and chiropractor. In addition, a massage is offered for a deductible of NOK 100 annually. Employees can book as many hours as they want. Schibsted has a large canteen, which accommodates 600–900 employees. Schibsted has its own intranet page where employees can view scheduled training hours, activities, promotions, offers, and they can sign up for activities. The website is based on Google technology, which makes it easy to connect to different devices. Internal sport competitions are also communicated through this website, as well as other general events and information related to health and the body. For informal communication about physical activities, the Slack platform is also used among active employees.

#### 3.2.3. Gjensidige

For more than 200 years, Gjensidige offers a wide range of insurance products to individuals, families and companies. The company’s vision aims to be a leading insurance company, both in the Nordic region and in the Baltics. The strategic focus is a high degree of customer orientation in combination with cost efficiency. Making sustainable choices and solutions are important prerequisites for long-term value creation [98]. The insurance company has 30–35 offices in Norway, with headquarters in Oslo. Gjensidige’s current group strategy for the next few years signals a focus on the dimensions of technology, urbanisation, climate and environment, regulations, and especially health. Gjensidige has approximately 1200 employees in two locations in Oslo. Everyone has access to the head office’s gym, which is referred to as Oslo’s best fitness facility among all the city´s companies.

‘We have a training area, which according to our partners in physio, is one of Oslo’s best company training facilities’. The changing rooms are equipped with 260 lockers that the employees can use both before and after a training session. The spinning room can accommodate 16 people and employees can train in a large gymnasium of 28 × 22 m, which is shared with the transport company VY. There are changing rooms with showers and an associated sauna for men and women and a large bicycle parking lot for those who cycle to work. Most of the physical activities are posted on Gjensidige’s intranet. In addition, employees receive information via notices on doors and boards, as well as email. Gjensidige uses hired instructors for both strength camps and yoga sessions weekly. The company offers flexible working hours. The informant from Gjensidige specified: ‘It means that you can go down and run a ride on the treadmill at 12 o´clock if you think it’s good for you. The time you spend you can catch up another time, for example the day after. We try to encourage and facilitate flexibility when it comes to timing’.

#### 3.2.4. Wilhelmsen

‘Our ambition is to shape the maritime industry. Through innovation, expertise, quality products, and services we are able to meet the challenges and needs of the world´s global fleet’. The Wilhelmsen Group is an international logistics and maritime company based in 74 different countries and employing over 21,000 employees. The group’s ambitions are based on innovation and expertise. Through global cooperation, the company focuses on a sustainable future. Wilhelmsen’s core values are customer experience, collaboration, learning and innovation. Wilhelmsen Norway is based at Lysaker, with over 300 employees. There, the company has a well-equipped strength room, a large gymnasium in the basement of the building with associated women’s and men’s wardrobes. Wilhelmsen hires SATS instructors several times a week, who, among other things, offer dance lessons in the form of salsa and swing. Employees have flexible working hours with a stamping system, so that the employees can organise their working hours and training sessions freely.

In the activity areas, there are several facilities such as a gym and strength room, as well as associated women’s and men’s wardrobes with closet space. Wilhemsen has a company agreement with SATS, which offers employees membership at discounted prices. Instructors from SATS are hired to lead yoga and body pump classes in the company’s own premises. A separate sports team (WIL) administers the training programs, the group sessions, and WIL receives funding for the health care program annually. For internal health communication, the company uses a Teams page with a separate portal where information about training programs, sports events and sales of sports equipment is published. WIL offers the following activities: Alpine skiing, cycling, golf, high and low, kayaking, sailing, floorball, football, yoga, and other group activities in the gym. In addition, the sports team is responsible for the management of the company cabins located on the coast and in the mountains, cultural events and the annual party. WIL has also organised self-defence courses and dance courses. Table 1 exhibits the main features of the case companies.

## 4. Results

RQ1: Are there any indications of a health-promoting performance culture among the Norwegian companies?

When asked directly if the companies use the term performance culture (used as a theme in the guide and a code in the analysis), it was clear that performance culture was used by managers, and it was a word that was mentioned in various contexts in the workplaces. The following quotes from three of the companies confirm the usage: ‘Yes, I can promise you that’. ‘Yes, absolutely’. ‘Yes, we do, it is accepted’. The informants explained that in situations when managers talk about the company’s goals as employee performance measures, the concept of performance culture is mentioned. The informants had a common impression that the employees did not seem to react negatively to the use of the term, and that it is a way of describing the culture. Findus is very much oriented towards goals, especially sales goals. ‘And employees are performing well, and the company is doing great. But we care about health and culture’. The Findus informant emphasised that it must be attractive to work in the company. An objective is that the employees should be involved in more than just the job. ‘We want them to have fun at work so that they stay for a long time’.

The informants were asked several questions related to health culture (based on the codes health objectives, health values, norms, rituals, and supportive health environment). The companies run physical activities during working days to break up the long working hours, which represent routines and norms. Findus practices walks during (lunch?) breaks and the employees take the initiative to do strength exercises as a break from the work task. ‘If I get into the plank pose for a minute when I get tired at 2 o´clock, then I can manage until four’, the Findus informant states. Gjensidige makes employees aware of how much time is wasted on meetings. The insurance company encourages short meetings, often standing, or more informal meetings in the canteen. The employees are offered short physical sessions as energy breaks in social zones of the company. Schibsted and Wilhelmsen emphasise that their meetings are executed traditionally. All companies offer various activity classes consisting of, for example, circle training and yoga. Wilhelmsen stands out here by having offered dance as a common fun physical activity. ‘We have had swing courses, and they were popular. Many signed up’.

Findus annually monitors the number of employees using its health insurance offers in addition to the sick leave, turnover, and how many employees are participating in the company agreement with the fitness centre SATS. ‘When it comes to how the physical offers affect the productivity of the company, we have no concrete figures, but the company is doing well’. Schibsted uses the platform Schibsted Life, which can provide an overview of the number of participants doing physical exercises, but systematic analyses are not carried out. Nevertheless, Schibsted expresses that the issue of value creation in relation to employee drivers is interesting to consider. ‘It sounds ideal, but we are not there now, no’. Gjensidige conducts monthly employee surveys that contain questions about salary, follow-up and physical conditions of its office space. The survey provides employees the opportunity to give feedback on facilities and programs. The insurance company does not assess employees’ health standard, however, the company reports low levels of absence due to sickness Wilhelmsen completes a health check of employees’ annually (blood test, vision/eye test, and hearing). The health check especially reveals the health condition of employees who smoke or suffer from cardiovascular disease. The company’s sickness absence is around two percent, which the company believes is the result of the good and varied offers of healthy and tasty food, attractive activities, and social events. Nevertheless, value creation as a consequence of the health care program is neither measured nor quantified.

RQ2: How and why do companies facilitate physical activity in the workplace?

A relevant question is if the health care programs and investments in a supportive health environment are anchored in an HR strategy, if leaders or managers participate, and how and why the facilitation of physical activity is done. The sub-themes here were HR strategy, leaders´ participation, organisation, programs and employee participation. Of the four case companies, only Findus has established the physical activity program as a direct consequence of a health care management thinking and an HR strategy through the program ‘Our Well Way’. ‘We are a team with three employees who run this together. As the Business Manager in sales, I am an ambassador and work closely with the HR Business Partner and a HR consultant’. The Findus informant states that health and wellness are linked to the company’s core values and the brand. Schibsted’s organisation of the physical activities, on the other hand, is based on volunteers. ‘We work voluntarily, and I do not get paid extra for it. The concept of health is not mentioned, but it is implicit in our values. We could probably have been better at getting it down’. Gjensidige has established a type of an internal organisation named ‘Gjensidige Sport’, which is led by volunteers and governed by a board. The program consists of 15–20 activity categories available for employees. The Gjensidige representative argues that ‘This way of organising is not based on a strategy directly. It is in accordance with our preventive HR measures, but it is not written down’. Wilhelmsen has a sports team organisation (WIL) that is managed by volunteers. ‘We have a varied offer, and it is up to the employees to use it. It is not directly in the strategy. But having a good time at work has always been a priority. We spend a lot of money on activities for the employees’.

The informant from Findus mentioned that the health care strategy combined with voluntary commitment contributed to more employees participating in more activities. The HR Business Partner emphasises that the management’s focus on health and well-being is important for motivating employees to participate in joint activities. This exemplifies that committed leaders can appear as role models, a factor of a health culture. Findus remarked that the commitment among some of the department managers could be stronger. ‘It is difficult to motivate and push others to participate when the manager does not participate’. The informant from Gjensidige stated that managers have different views on whether training and health should be given priority. ‘Leaders are different. Some would say it’s okay, others would wrinkle their noses’.

Findus explains that through the ‘Our Well Way’ program and the three focus areas a year, the goal is to strengthen the knowledge about health benefits of among others physical activity and a healthy diet. External experts contribute with presentations and speeches. Findus keeps track of who is taking part in the program and activities. It seems that those who are physically active in their private sphere are those who participate in physical activities at work. The informant from Wilhelmsen indicates the same tendency. ‘It is difficult to reach those who do not do physical exercises at all. Employees who exercise at home, do physical training at work. Our experience after the lock down in March 2020, due to the Covid 19 virus, is that more employees have joined our physical activities’. The informant from Wilhelmsen explains that offering low-threshold activities is important in order to motivate as many employees as possible. ‘That’s why we started dancing lessons, as we thought salsa was fun’. This offer was described as more popular than other traditional physical activities. Nevertheless, the informant believes that there is a limited effect of low-threshold activities. The informant from Gjensidige demonstrates uncertainty about effects due to the lack of a measurements system. ‘There are employees who find everything about physical activity and health uninteresting, while others exercise several times a week at work and use their bicycle to work’.

RQ3: What kind of facilities and structural factors are offered to employees?

The sub-topics/themes of this question were the related supportive health environment and specifically the offers/benefits, facilities, programs, and communication/health technology. Schibsted, Gjensidige, and Wilhemsen offer good facilities with gymnasiums, strength rooms, spinning halls, bicycle racks and changing rooms. Findus will be moving to a new building during the summer with similar facilities. Schibsted, Gjensidige and Wilhemsen have agreements with the fitness centre SATS about hiring instructors several times a week to offer physical activity sessions to their employees (e.g., strength camps and yoga sessions). WIL (Wilhemsen) has also offered successful dance lessons (salsa and swing). The companies provide a healthy and varied diet in their canteens, free health insurance, easy access to physical treatment, massage, and flexible working and training hours. In the case of Gjensidige, we noticed that the informant specified: ‘It means that you can go down and run on the treadmill at 12 o´clock if you think it’s good for you. The minutes you spend you can catch up another time, for example the day after’. Concerning Schibsted’s health insurance, employees can download a mobile app that can be used to book appointments with a doctor, physiotherapist and chiropractor. Free fruits, green meals and vegetarian days are common among the case companies.

As part of structural factors, the informants responded to questions about how their communication functions. Findus has its own intranet, but it is not used in regard with the promotion of physical activities. The informant explained that they may be able to offer an app when the company has changed location. A test with excel sheets encouraging employees to register training sessions was not successful. Schibsted has a concept for the entire group called Schibsted Life (part of the intranet), which is also used to disseminate information about social gatherings and campaigns. For informal dialogue between physically active employees, Slack is used. ‘Gjensidige displays timetables for activities using a calendar application via Outlook’. Information in paper format about exercise programs and health issues is sometimes posted on the doors in the office space. Wilhelmsen uses a portal via Microsoft Teams. ‘There you go in and find what you need’. However, the company does not have a solution for exercise registration or information about health and diet.

## 5. Discussion

Businesses and organisations depend on good performance through the delivery of products, services, and experiences in order to produce positive brand associations and good financial results over time. As my theorising and the corresponding summarised model in Figure 1 illustrate, I argue that a sound level of employees’ mental and physical health and a health-promoting performance culture are vital prerequisites for employee and organisational performance. Based on this theory, the research questions about a health-promoting performance culture (RQ1), the reasons for implementing employee health care programs and how (RQ2), facilities/structural factors (RQ3), and the collected qualitative data, it seems that the recommendations from the theory have yet to be implemented in many of these companies´ practices.

Related to RQ1, common to all the case companies is that they identify their organisations as having performance cultures. There are indications of such a culture, as the companies claim to be goal-oriented, and trust, freedom, and autonomy [44] are reflected in the flexible working hours. In addition, Findus admits being very focused on sales goals. The canteens offer free fruit, healthy food, vegetarian days and meals, which contribute to employee health [5]. Offering healthy and tasty canteen food and worksite physical activities are indicators that the companies are moving toward a health culture [15]. Based on the companies´ values, the companies promote physical activities and facilities with a high standard of supportive structural factors [50]. The health programs consist of various group activities such as yoga, circuit training, dancing and a number of team sports. Based on these findings, it can be argued that three of the four companies have developed a health-promoting environment [4,99] but not yet a health culture [15]. However, Findus can be argued to have a health culture [50] due to the link to the HR strategy, the health values, the ‘Our Well Way’ program, the three focus areas a year, and the goal to improve employees´ understanding of health benefits of physical activity and a healthy diet [5]. There are also indicators that Findus is in a process of developing a health-promoting performance culture, as the company is goal-oriented and has implemented the health program ‘Our Well Way’.

From the cases, as well as the theory, support is found for the theory that that unifying physical activity events and health care programs can contribute to social interaction [19], improve employee relations [70], strengthen health [4,5], and increased fun at work [69], which in turn, can lead to the development of a health-promoting performance culture. Participation, commitment, engagement and partnerships [100] such as those the companies have with the fitness centre SATS, are probably, as the theory also suggests, important factors in stimulating employee physical activity. However, employees going to the fitness centres on an individual basis will not contribute to the development of a health culture, improve fun at work, increase socialisation, and relationships, or strengthen the sense of community. Facilitating and promoting effective health care programs and common activities with leader participation, as Findus is practicing, is an important finding, new to the literature. Such a practice is likely to be an effective HR strategy [53].

When it comes to participation and how the activities are organised (RQ2), it was claimed that those who train consistently at work are the same as those who already train home. These employees are also represented by the volunteers who organize the workplace physical activity. Findus differs somewhat from the other case companies because the employees of the responsible team, also consisting of managers, appear as good role models and represent participatory leadership [38,40] and supportive leadership [24]. The case and the interview show that leaders take part in and are responsible for planning and implementing the ‘Our Well Way’ program. The manufacturer of frozen foods is the only company of four that has developed a comprehensive health care program. The annual program and value wheel and a systematic approach to communicating the health benefits of physical activity indicate the company is developing a health culture [15]. The findings show that all companies use the term performance culture with a clear goal orientation. There are indications that possible negative connotations linked to performance culture are toned down when employee-based health care measures like working flexible hours and the offering of healthy lunches are promoted.

According to Golaszewski et al. [15], individual factors constitute employees’ health knowledge, attitudes, skills and values that can have an impact on a company’s health culture. Concerning RQ3, the companies did not have a separate health portal, as a part of a supportive health environment, similar to the one described in the Aker case [78]. It was also lacking a data management system as Golaszewski et al. [15] recommend, which enables a systematic analysis of the effects of physical activity at work. Based on the theoretical links between key constructs reviewed above, it should be possible to link employee data on mental and physical health, fun at work, job satisfaction, motivation, commitment, engagement and organisational culture to absences due to sickness and related costs. All the informants seemed interested in using a measurement system which could reveal investment efforts and their effects. The companies in this study have invested significantly in supportive structures [50] to facilitate employee physical activity. A portal could more effectively promote health values and increase employees’ health knowledge and attitudes. For example, information about seminars with health topics similar to what Findus has implemented could be linked to texts about effects of a healthy lifestyle, diet and physical exercise. A portal can also contain suggestions on training programs and diet. Thus, a portal, as one example of a supportive structural factor, can aid the communication of the wellness program [77] and the implementation of a value-based health care program aiming at improving employees’ physical activity level and skills. As mentioned previously, Price, Bray, and Brown [63] recommend that canteens should offer mobile apps that can provide information about the nutrition of the canteen food. The communication function should be given priority, as it is a challenge to motivate and convince the majority of employees of the health benefits of physical exercise and a healthy diet, and the primary target group of employers should be those who are not as physically active.

## 6. Conclusions

The interdisciplinary perspective of the literature review is based on different directions within organisational theory, organisational psychology, psychology, health care management, and health promotion. A main argument is that a health-promoting performance culture and individual HR drivers must be nurtured and developed to stimulate employees and the organisation to perform. After having theorised, collected and analysed qualitative data to answer RQ1, there are indications that the findings support the theory of a health-promoting performance culture, and individual HR drivers aid organisational performance. The theory suggests that a collective health boost can affect, for example, sick leave and other employee-drivers like fun at work, job satisfaction, motivation, commitment, and engagement, which can move organisational performance in a positive direction. However, the companies do not seem to have established such a reasoning when facilitating for employee physical activity.

As this is a conceptual and exploratory study, Figure 2 is a result of both theories and findings from the cases and interviews. An important finding is that a key success factor for developing a health-promoting performance culture is the involvement and participation of leaders in employee physical activity. This was exemplified in the Findus case (see the health care strategy in Figure 2). Similarly, as Findus is emphasising, the health-promoting programs should aim at strengthening employees´ health knowledge of the benefits of a healthy lifestyle through seminars that are broad in scope and relevant (see health-promoting seminar and health portal in Figure 2).

The findings related to RQ2 and RQ3 about facilities, structural factors, and programs aiming to develop employees´ mental and physical health, for which the HR department must take responsibility (like Findus), are in line with existing theories. New to the theory is that companies invite external instructors (from SATS) to run physical activities at the workplaces. Such an idea can be expanded to strengthen the supportive health environment. Instead of using external partners similar to the fitness chain SATS and various treatment institutions, a supportive health team consisting of a coach/trainer, nutrition expert, and physiotherapist/masseur can be part of the HR department (see Figure 2). For example, the health team can encourage and motivate employees to do health-promoting meeting activities such as ‘walking meetings’ or ‘jogging meetings’ as part of the health-promoting program. As Figure 2 shows, by establishing a health care strategy (health goals, supportive health environment, and participative leaders), health care management (programs, partnerships, effective communication), implementation, and health technology, a health-promoting performance culture and important values will be created. More specifically, seminars on health-related topics and a separate health portal with relevant functions (registration of participation, registration of physical exercises), content (health information, exercise programs and diet suggestions), and a feedback system will aid the continuous development of a health-promoting performance culture. 

## 7. Implications

This article deals with Norwegian cases in a Norwegian context. The main message in the Norwegian Public Health Act and the national strategy HealthCare21 is that we must assess the health risks in everything we do and initiate preventive measures continuously. Employers can implement health-promoting programs and support the positive effects of individuals’ own free, life-prolonging ‘treatments’ like physical activity, social interaction, non-smoking, and a healthy diet. Implementation of such health care strategies should be researched in the future. I also suggest that new occupational safety and health (OSH) directives related to dietary, mental and physical health should require employers to invest in supportive health environments and physical activity programs targeting employees. Thus, a collective, nationwide effort can contribute to a health boost among the employed part of the population, and thus to improved public health. The national arena can be broadened to a European context. Pan-European quantitative research should be completed to understand the accumulative effects of several companies´ health care strategies, suggested under value creation in Figure 2.

## Figures and Tables

**Figure 1 ijerph-17-09164-f001:**
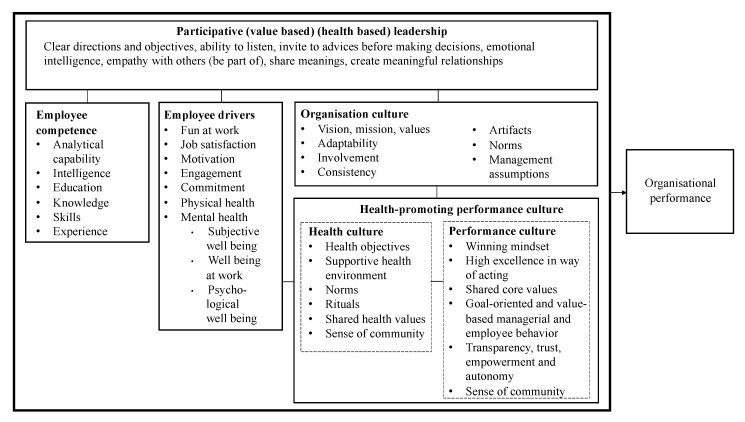
A health-promoting performance culture and individual HR drivers of organisational performance.

**Figure 2 ijerph-17-09164-f002:**
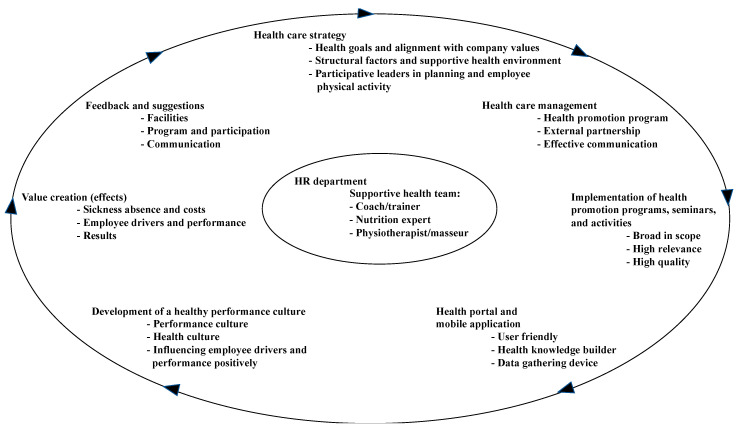
The implementation of a health-promoting performance culture.

**Table 1 ijerph-17-09164-t001:** Main features of the case companies.

Company/ Features	Findus	Schibsted	Gjensidige	Wilhemsen
Number of employees and participants of employee physical activity	45 people work in the main office where the facilities are located. Around 10–15 employees participate in common activities on a regular basis.	In the main office, where the gym is, there are 1200 employees and about 450 in a nearby location. There is room for 35 participants per training hour and may be 150–200 use the facilities during a week. Around 150 people are logged in during home based online training sessions.	About 150 employees use the facilities daily (out of 820 in the main Oslo-office and 350 in a nearby location).	Of 300 employees in the main office, about 30–40 employees use the facilities and activity offers on a daily basis.
Physical facilities	Sharing facilities with other companies. Fitness and spinning rooms. Locker rooms and showers.	A large fitness centre, which includes two activity rooms	Large gymnasium, spinning room, sauna, locker rooms, showers, bicycle parking	Gymnasium and strength room. Wardrobes with facilities.
Health promoting programs and organising	The HR-based ‘Our Well Way’.	Various common activities like yoga organised by volunteers.	Various sessions organised by volunteers or professional instructors.	Offers several sport team activities in the gym organised by the sport team (volunteers) or professional instructors.
Canteen and diet	Green weekdays with meat-free meals, free fruits and canteen food with a healthy and varied diet.	Promotion of different themes on a weekly or daily basis such as green days. All employees receive free water, coffee and fruits.	Own canteen personnel put strict requirements on the canteen food. Healthy food and green meals.	Company chefs and canteen staff emphasizing serving healthy, green and nutritious food. Vegetarian days.

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
