# Peer review of "Towards a HR Framework for Developing a Health-Promoting Performance Culture at Work: A Norwegian Health Care Management Case Study"

_ijerph, 2020, doi:10.3390/ijerph17249164_

Round 1

Reviewer 1 Report

  • The topic presented in the article is important and current.
  • Some relations, which presents figure 1 are not clear, for instance what is the relation between "participative leadership" and "intelligence" or "analytical capability", how they are influenced by the leadership (such influence is presented)?
  • In the article we can find the term "healthy performance culture" as the sum of a "health culture" and a "performance culture". In my opinion all these categories should be defined more precisely: separately and together.
  • I recommend also to change the figure 1. If the "healthy performance culture" is the "marriage" of the "health culture" and "performance culture" as the author notes, he also separately should present the determinants of the "health culture" and " performance culture", including interactions among them.

Author Response

Dear Reviewer 1.

Thank you for being critical to the key constructs in the manuscripts, their relations and Figure 1.

  1. I have tried to explain the relations better.
  2. I have explained/defined health culture, performance culture and based on the theorizing proposed a definition of a healthy performance culture new to the health management literature.
  3. I have changed Figure 1 substantially based on your review. Now, it is much more detailed and to the point including all the key constructs.

Best regards,

Reviewer 2 Report

This paper presents a case study-based qualitative research on corporate health promotion initiatives and their impact on employees and their corporate culture, and performance. The efforts of this study are very significant because improving the health of employees is expected to be involved in reducing the overall health care costs of the country and also in improving corporate performance.

However, the structure of the article as an academic paper is a major challenge.

In particular, the aim of this study is to develop the construct of HEALTHY PERFORMANCE CULTURE, but not enough logic has been proposed to develop such a new construct.

In this paper, there is an explanation for the alignment of performance-related and healthy culture-related theories in business literature. If that, it is necessary to explicitly explain how such a theory is to be integrated. It is then necessary to discuss what data analysis can be done to validate the new construct that is based on existing theories. For example, based on the theoretical background of the two disciplines, the hypotheses of this study could be presented in terms of how healthy performance culture is developed here.

With regard to the methodology, due to the study was qualitative with thematic analysis, there can be the coding results based on the interview data with each company. But, there is no systematic explanation about the code, such as how the statements of the four companies are explained for the topics used here. The current description gives the impression that only convenient statements from informants are picked up and used to develop the logic.

In addition, the final HR framework presented should be based on the results of validating the research questions presented in this paper.

It is necessary to discuss how research questions linked to the theoretical background have been solved and the theories modified by the data obtained in this study, and then present the practical implication as an HR framework that incorporates the new findings.

Also, as a basis for the authors' arguments in the Discussion section, please make available references to what codes and actual statements of informants based on the qualitative analysis.

Here are the details of the comments tied to the line numbers

------------

8: What does the acronym (FHI) stand for?
10: Just to confirm, please refer to the journal guidelines or other papers in this journal for the currency symbols.
25: Please makes it clear that they are the name of the company.
28: In the abstract, please also indicate what this paper has acquired as a new finding.
35: Check the sentence structure again.
64: For example, a structure of a standard business-oriented paper provides a research question based on the theoretical background and explains what data will be acquired as a means to solve it.
219: This is where RQ1 comes in, but the subsection before that is just an introduction to the existing literature. Please clarify the logical relationship between the theoretical background of the study and the derivation of the RQs/Hypotheses.
242: A typo?
258: Since the construct of healthy performance culture is not defined in the first place, we can't answer this question without clarifying it.
300: What is the reason for explaining this section?
530: In the discussion, first explain how the analysis of the data you obtained allowed you to present answers to the research questions you set.
584: You have given Berry's (2010) criteria here, but I am not sure why you have brought this perspective here and the details of how these criteria relate to the qualitative results obtained in this study. I think the figure 2 is important, so please explain in detail.
596: A typo?
597: Check for consistency with the purpose of the introduction section.
598: I'm not sure where the subject starts.
603: Has the analysis in terms of firm performance been discussed in this paper? Please check the correspondence of the arguments made in this here and the facts that support them.
609: What is the basis for this statement?
611: Before giving Figure 2, please specify how the construct of healthy performance culture linked to existing theories is validated by data obtained.

------------

Author Response

Dear reviewer 2.

Thank you for a thorough and encouraging feedback.

Alle changes are colored red.

  1. I have reorganized the theory, strengthened the argumentation and logic, defined the types of cultures, rephrase three research question, which I have followed up under findings and discussion.
  2. I have added missing references, deleted some.
  3. The abstract is partially reformulated and some findings are added.
  4. The introduction is strengthened with a few point to underscore the situation and starting point in terms of health challenges.
  5. I have re-phrased the title and the key constructs. It is mainly about a healthy performance culture and individual HR drivers as forces behind organizational performance. Thus, figure 1 is showing the constructs in a clearer and more detailed way.
  6. Ref Figure 1 and employee competence (education/intelligence). It is linked to performance culture and responsibility of leaders (under participative leadership).
  7. I have re-phrased the research questions (three of them). They are more open now and better aligned with the conceptual and exploratory design. 
  8. The findings are organized in accordance with the RQ1-3 so the paper seems more structured from the theory, to the finding and discussion.
  9. Method and codes. I have underlined/added the codes at the beginning of the RQs under Results. It should be easier now to relate findings to the theory. However, I am not sure if I understood this point correctly. However, the themes from the guide and codes from the methodology correspond better now with the themes listed un der Results and the RQs.
  10. I have made it clearer that this a conceptual and exploratory study, and that the empirical findings support the theory that is developed. Also, I have stressed a few important findings that it not covered in present research.
  11. This also applies to the final framework, which is based on both theory and findings.
  12. As mentioned, I have split the research question and made three of them. Thus, the theory section is easier to follow. When the RQs  are organized in the same manner (with codes), I think it should be easy to see the link between theory, results and discussion (which is restructured and amended more in line with the results and RQs). Besides, it is clearer that the cases to some extent mirror the theory. And there are some valuable practices among the companies that are built into the final framework. Also, the theory and findings must be seen in a larger context and the idea that the accumulative value of employer organizing for employee physical activity may be great.
  13. The challenges linked to the specific lines are resolved (see text in red).

Round 2

Reviewer 2 Report

The framework (Figure 2) is presented as an output from this study, but it is not easy to read how the author integrated the individual results. For instance, could you also answer the research questions from the overall perspective of the four cases?

The discussion or conclusion section is also not easy to clearly read the practical and theoretical implications of this study. Please underline these points explicitly as well.

Below are detailed comments with line numbers.

----------

11: What is the unit of cost?
23: Regarding the result, isn't it ambiguous in ``may''? Could you articulate the interpretation you can make based on the data obtained in this study?
41: Duplicate punctuation.
94: Please unify the indentation throughout the paper.
96: Why did you focus on these theories in this study?
132: Is there one author?
138: Check the grammar.
149: A period is missing.
153: Is this statement also a claim by Gurt and Elke?
286: I'm not sure why the structure in Figure 1 organized like this, such as the meaning of the lines. So please describe the figure more clearly or add an explanation in the main text.
325: Please also explain the reason why you focused on these four companies in your data collection.
479: This section contains some duplicate statements from the previous section. Could you summarize and tidy-up the contents?
612: Extra period

----------------

Author Response

A.

The framework (Figure 2) is presented as an output from this study, but it is not easy to read how the author integrated the individual results. For instance, could you also answer the research questions from the overall perspective of the four cases?

The discussion or conclusion section is also not easy to clearly read the practical and theoretical implications of this study. Please underline these points explicitly as well.

Author:

Discussion

I have re-structured this section and linked findings/dsicussion to the RQs.

Conclusion

I have tried to clarify and I have brought in the research question to make a clearer link between theories and findings. And also, the contribution - what is from the theory and what is found from the data.

-"As this is a conceptual and exploratory study, Figure 2 is a result of theories and findings from the cases/interviews". See from line 679 an on.

See also from line 686 and on. I have tried to link the RQs, Figure 2 and findings.

B.

11: What is the unit of cost?: Done: "1,75 billion"

23: Regarding the result, isn't it ambiguous in ``may''? Could you articulate the interpretation you can make based on the data obtained in this study?

Done: "are the results" 41: Duplicate punctuation.Done
94: Please unify the indentation throughout the paper.Done
96: Why did you focus on these theories in this study? Done. I have added an introductory sentence and re-phrased: "Below, it is theorized the relations between the key theoretical terms of this study".
132: Is there one author? Done: also added to the reference list (Spector & Lane, 2007).
138: Check the grammar. Lines 134-143 - de Waal argued, I have re-phrased the definition of performance culture by the help of an English colleague.
Otherwise, I am not sure of any grammatical errors.
149: A period is missing. Done. Now 151.
153: Is this statement also a claim by Gurt and Elke? Done and re-phrased (154-155)
286: I'm not sure why the structure in Figure 1 organized like this, such as the meaning of the lines. So please describe the figure more clearly or add an explanation in the main text. Done. Good point. The lines between culture and employee competence and between competence and employee drivers are omitted as these relations are not clearly supported in the theory. I added a sentence in the beginning of the section (line 94) and a summary explaining Figure 1. (see lines 276-283).
325: Please also explain the reason why you focused on these four companies in your data collection. Done. One explanatory sentence is added.
479: This section contains some duplicate statements from the previous section. Could you summarize and tidy-up the contents? I am not quite sure about this one. The section on Cases ends now on line 480. Line/sentence 484 is changed and mentions only performance culture, which is dealt with in that paragraph. The next paragraph deals with health culture. I do not see any duplicate statements.
612: Extra period: Done  

C.   Part 3.2 The setting and background is omitted. I thought this was done in the last submission (after the first revision) as part of it was placed at the beginning of Introduction. It is ok now.

D.    I have changed the definition of healthy performance culture slightly (see line 182) through the help of an English colleague.

E. I have slightly re-structured the past paragraph of the Conclusion and added a sentence (in red).
